# Rydberg arrays offer a universal resource for measurement-based quantum computations

Valentin Crépel

*Department of Physics, Massachusetts Institute of Technology,*
*77 Massachusetts Avenue, Cambridge, MA, USA*

We show that one of the quantum entangled state recently realized in a Rydberg quantum simulator [Semeghini *et al.*, Science 374, 6572 (2021) [1]] offers a sufficient resource for universal measurement-based quantum computation. In particular, we provide explicit measurement sequences that implement a universal set of gates on the encoded logical qubits. Upon successful experimental realization, the proposed sequences would promote the Rydberg simulators to the first universal quantum computers relying on the measurement-based model of quantum computation.

## I. INTRODUCTION

Quantum simulators are controlled quantum devices that can be used to simulate other quantum systems, providing the means to study a class of problems currently intractable on classical computers [2–4]. Being special purpose devices, they are often contrasted with general-purpose universal quantum computers [5, 6], which are – in principle – capable of solving any quantum problems, including those addressed by simulators. This apparent, but loosely stated, inclusion relation is however misleading, as it is in fact an equivalence.

Indeed, certain quantum entangled states, once simulated, provide sufficient resources for the realization of any quantum algorithm using the model of measurement-based quantum computation (MBQC) [7–10], wherein measurements on individual constituents of the entangled state drive the algorithm. Within the MBQC paradigm, quantum simulators realizing those computationally universal states (CUSs) are therefore equivalent to universal quantum computers [11, 12].

The standard example of such CUS, and the first identified, is the cluster state on the square lattice [13, 14]. Since then, several other two dimensional states – such as graph states [15, 16], the tricluster state [17], modified toric code states [18], and Affleck-Kennedy-Lieb-Tasaki states [19, 20] – have been diagnosed as CUSs. Intriguingly, all these candidates for universal MBQC share a common feature, they are two-dimensional symmetry-protected topologically ordered states (SPTOSs) [21–25]. The coincidence between CUS and SPTOS was further grounded numerically in certain models [26, 27]. Using this phenomenological observation, the search for the first quantum simulator to be promoted to the rank of universal quantum computer might benefit from a focus on those realizing two-dimensional SPTOS.

Recently, a two-dimensional state with $\mathbb{Z}_2$ topological order has been realized in Rydberg arrays [1, 28], one of the most promising quantum simulator for the study of many-body systems with short-range interactions [29–31]. In this article, we show that this SPTOS exhibits universal computational power within the MBQC paradigm. To achieve this goal, we first model it as a pair-entangled projected pair state (PEPS) [32], and then provide explicit and experimentally realistic measurement sequences implementing a universal set of quantum gates. If successfully realized, Rydberg arrays could be engineered into universal quantum computers relying on MBQC, a paradigmatic shift from the circuit model implemented, *e.g.* with superconducting qubits [33, 34].

Our findings call for an extensive search of UCS in Rydberg arrays, tracking those offering the simplest experimental sequences for the implementation of a universal gate set. Experimental imperfections weakening the reliability of MBQC, such as non-adiabaticity or dephasing due to long-range interactions, should also be identified and listed, together with ways to correct for them, *i.e.* through pulse-engineering.

## II. REQUIRED EXPERIMENTAL RESOURCES

Let us first describe the SPTOS realized in Ref. [1] and characterize the types of measurement available in typical Rydberg array experiments. These two types of resources will later serve as the necessary building blocks of our MBQC scheme.

### A. $\mathbb{Z}_2$ SPTOS in a Rydberg simulator

The Rydberg array of Ref. [1] consists of atoms individually trapped in optical tweezers [35–37], and positioned on a Ruby lattice, *i.e.* on the links of a Kagome lattice (see Fig. 1a). These atoms are initially prepared in an electronic ground state $|g\rangle$, which is laser-coupled to a Rydberg state $|e\rangle$ with a Rabi frequency $\Omega$ and a detuning $\Delta$, leading to the on-site Hamiltonian

$$\mathcal{H} = \frac{1}{2} \left( \Omega |e\rangle\langle g| + \Omega^* |g\rangle\langle e| \right) - \Delta n, \quad n = |e\rangle\langle e|. \quad (1)$$

Due to their high polarizability, atoms in the Rydberg states strongly interact through the strong van der Waals interaction $V(d) = C_6/d^6$, with $d$ the interatomic distance [30]. As a result, all atoms within a distance $r < R_b = (C_6/\Omega)^{1/6}$ of an atom in $|e\rangle$ are brought far off resonance from the laser field. This effect is well captured

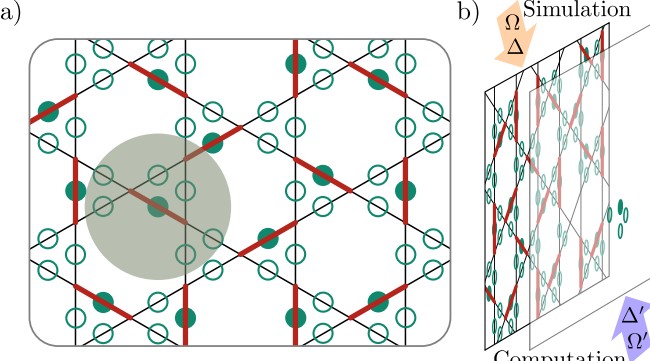

FIG. 1. a) State consistent with the blockade condition (grey circle) at filling $\langle n \rangle = 1/4$, the equivalence with dimer is made explicit by coloring the links of the Kagome lattice on which Rydberg excitations lie. b) Physical decoupling of the simulation and computation (or measurement) plane allow for pulses with arbitrary driving parameters $(\Omega', \Delta')$.

by the so-called blockade radius model, in which the system's dynamics is approximated by Eq. 1 projected to the subspace where no pairs of excitations distant by less than $R_b$ are present.

In Ref. [1], the Rabi frequency and detuning have been set such that $2a < R_b < \sqrt{7}a$, $a$ being the Ruby lattice constant, and $\langle n \rangle = 1/4$. The first condition prevents adjacent links of the Kagome lattice to simultaneously host an atom in $|e\rangle$, while the second requires that at least one of the four links surrounding each vertex be in the Rydberg state. As sketched in Fig. 1a, these conditions restrict many-body configurations to dimer coverings of the Kagome lattice, where the position of dimers is given by the excited atoms [38].

After appropriate time evolution under the blockade radius model Hamiltonian, the state $|\Phi\rangle$ realized with the previous parameters is very well approximated by a dimer state, i.e. an equal superposition of all existing dimer coverings. The overlap between the dynamically prepared and dimer states was estimated to reach as high as 99% for 48 atoms under realistic experimental conditions [28]. In the following, we shall therefore approximate $|\Phi\rangle$ by the Kagome dimer state. This state has been thoroughly studied in the literature [39–41] and is known to host $\mathbb{Z}_2$ topological order [42–44].

The first goal of this paper is to explore the the computational power of $|\Phi\rangle$ within MBQC, and prove that it is a CUS. Our second aim is to show that universal MBQC is within experimental reach. The measurements sequences proposed in this article should therefore remain realistically implementable in current Rydberg array experiments.

## B. Available measurements

To meet this last criterion, we shall only rely on fluorescence measurement of the Rydberg excitation number at each site, potentially preceded by a 'pulse'. A pulse is defined as the free evolution of a few atom cluster decoupled from the rest of the lattice under the Rydberg blockade model Hamiltonian. Note that pulses might require different laser parameters $(\Omega', \Delta')$ than those used to stabilize the dimer state. One possible setup allowing for this discrepancy is depicted in Fig. 1b: atomic clusters are moved out of the simulation plane into a computation plane, where a different laser is shone to produce the desired pulse and where the final fluorescence measurement takes place [48].

In the next section, we will combine three elementary pulses to form a universal set of gate on the encoded logical qubits: (I) The first one involves a two-atom cluster, evolved under Eq. 1 for a time $\pi/\Delta$, with the detuning tuned such that $\Delta/\Omega = \sqrt{2/3}$ [49]. The Hamiltonian in the two-atom Rydberg blockaded subspace $\{ \succ, \succ, \succ \}$ is easily exponentiated to give

$$e^{-i\mathcal{H}\tau} = \begin{bmatrix} -1 & 0 & 0 \\ 0 & q & q^* \\ 0 & q^* & q \end{bmatrix}, \quad q = -\frac{1+i}{2}. \tag{2}$$

The measurement of the Rydberg excitation position $P = \mathrm{diag}(0, 1, -1)$ after this time evolution becomes equivalent to the measurement of

$$e^{i\mathcal{H}\tau} P e^{-i\mathcal{H}\tau} = \begin{bmatrix} 0 & 0 & 0 \\ 0 & 0 & 1 \\ 0 & 1 & 0 \end{bmatrix}, \tag{3}$$

on the original lattice, and projects onto the states $\succ$, $(\succ + \succ)$ and $(\succ - \succ)$ when $p = 0, 1, -1$ is measured, respectively. (II) The second pulse isolates three atoms within a blockade radius for a time $4\pi/(3\sqrt{3}\Omega)$ before fluorescence measurement. A similar analysis on the blockaded Hilbert space $\{ \triangleright, \triangleright, \triangleright, \triangleright \}$ shows that the state of the atomic cluster is now projected on $\frac{1}{2}(\triangleright - p_\uparrow p_\downarrow \triangleright - x p_\uparrow \triangleright - x p_\downarrow \triangleright)$, where $x/p_\downarrow/p_\uparrow$ are equal to one when the left/bottom right/top right link of the triangle is in $|e\rangle$, and to minus one otherwise [50]. (III) Finally, a far detuned laser field $(\Delta \gg \Omega)$ can be applied for a time $t$ to imprint a variable phase shift $\varphi = \Delta t$ on the $|e\rangle$ state of individual atoms.

In the following paragraphs, we show how MBQC can be implemented by combination of these three measurement pulses on $|\Phi\rangle$.

## III. UNIVERSAL MQBC

Our scheme for MBQC using $|\Phi\rangle$ as a resource follows the principle of computation in correlation space [16, 18, 47]. More precisely, we write the dimer state as a PEPS,

and encode the logical qubits in the auxiliary space of the PEPS tensor. Quantum information processing is performed by pulse measurements that act as special contractions of the tensor's physical space. The residual action of these contracted tensors on the logical qubits executes a desired set of gates.

### A. Tensor network description

To define our logical qubits and to make the MBQC scheme transparent, we introduce one possible tensor network representation of the dimer state

$$|\Phi\rangle = \qquad , \qquad (4)$$

that uses a PEPS tensor of bond dimension equal to two, and a physical space gathering six atoms forming a bowtie. This tensor only has eight non-zero elements, all equal to one, that can be depicted as

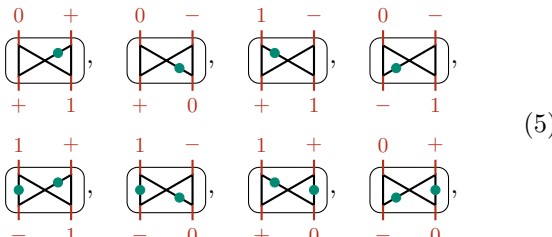

$$(5)$$

with filled circle representing atoms in $|e\rangle$, and where we have used the notation $|\pm\rangle = (|0\rangle \pm |1\rangle)/\sqrt{2}$ in the auxiliary space. The two upper auxiliary qubits are equal to ones or pluses if the vertex closest to them is covered by a dimer, and to zero or minus otherwise. The opposite is true for the two lower auxiliary qubits, such that, upon contraction, only configurations with all vertices covered by one and exactly one dimer acquire a non-zero weight. Since all coefficients in Eq. 5 are equal to one, this weight also is, and Eq. 4 indeed represents the dimer state. Note that, because we kept the physical degrees of freedom of the tensors on the atomic sites rather than on the vertices of the Kagome lattice, our construction differs from, but is equivalent to, earlier PEPS descriptions of dimer states [45, 46].

Logical qubits are encoded into the auxiliary space of the tensors, and form computational wires (red vertical lines in Eq. 4). In the remainder of the discussion, we provide an explicit way to devise a universal set of gates on these logical qubits that solely rely on the pulses (I-III) given above.

### B. Wire decoupling

First, logical qubits should be able to pass through the PEPS tensors without being altered when no gate is applied. In other words, the two incoming logical qubits of each tensor should be decoupled in that situation. The following procedure allows to do so: the right and leftmost atoms of the bowtie are first isolated and measured in the $|g\rangle \pm |e\rangle$ basis (*i.e.* fluorescence after a $\pi/2$-pulse); then, a $\pi/2$ phase shift is imprinted on the two bottom atoms of the bowtie's central cross (III); finally, the atoms of the cross are gathered two-by-two vertically, each within its own blockade radius, and pulse (I) is performed.

Our above analysis shows that this sequence projects the PEPS' physical state onto one of the states

$$\begin{cases} \left(|+x_L\rangle\right) \times \left(\searrow + p_L \nearrow\right) \times \lessgtr \times \left(|+x_R\rangle\right) & \text{if: } p_R = 0 \\ \left(|+x_L\rangle\right) \times \gtrless \times \left(\nwarrow + p_R \swarrow\right) \times \left(|+x_R\rangle\right) & \text{if: } p_L = 0 \end{cases},$$

$$(6)$$

where $x_L, x_R = \pm$ are the measurement results from the right/left-most atoms, and $p_L, p_R$ are those of the left and right two-atoms clusters, as defined after Eq. 3. Using Eq. 5, we can then compute the action that this projection, written as $D_0(x_L, p_L, p_R, x_R)$, has on the incoming logical qubit

$$\boxed{D_0(x_L, p_L, p_R, x_R)} \equiv \begin{array}{cc} \boxed{H} & \boxed{H} \\ \boxed{Z^{p_R}} & \boxed{Z^{p_R + x_R p_L}} \\ \boxed{X^{p_L + x_L p_R}} & \boxed{X^{p_L}} \end{array}. \quad (7)$$

The two last lines are irrelevant Pauli errors that are inevitable within MBQC, and can be corrected at the end of the computation. We observe that the computational wires are indeed decoupled by $D_0$, up to Hadamard gates that will cancel out with those arising in adjacent PEPS tensors (see below).

### C. Single qubit operations

A small modification of the previous scheme allows to implement single qubit rotations along each of the incoming computation wires. More precisely, if the leftmost atom of the bowtie is measured in $|g\rangle \pm e^{i\varphi}|e\rangle$ using (III), the gate implemented changes as follows

$$\boxed{D_\varphi} \equiv \begin{array}{cc} \boxed{H} & \boxed{H} \\ \boxed{G_\varphi^{|p_R}} & \boxed{Z^{p_R + x_R p_L}} \\ \boxed{X^{p_L + x_L p_R}} & \boxed{X^{p_L}} \end{array}, \quad G_\varphi = ZR_x(\varphi),$$

$$(8)$$

where $R_x$ denotes a rotation around the $x$-axis on the logical qubit's Bloch sphere. The conditional rotation $G_\varphi$ is similarly performed on the right computational wire when the rightmost atom of the bowtie is phase-shifted instead of the leftmost one.

While this rotation is only applied when $p_R \neq 0$, we can try to implement it on successive PEPS tensor along the computational wire until a non-zero $p_R$ heralds successful rotation of the qubit. We now show that the probability to measure $p_R = 0$ on $n$ successive tensors is $K/2^{n-1}$, with $K$ constant, ensuring the quasi-certain application of the rotation $G_\varphi$ with polynomial circuit depth using the above iterative method.

To prove this statement, consider the following situation

$$\text{(9)}$$

and assume that we want to apply a rotation $G_\varphi$ on the central logical wire but have measured $p_R = 0$ on the upper bowtie. We thus apply a phase shift on the rightmost atom of the lower bowtie, and would like to know the probability of success ($p_L \neq 0$). The measurement $p_R = 0$ has restricted $|\Phi\rangle$ to an equal superposition of dimer covering in which either atom 1 or atom 2 is excited. Suppose it is atom 1, then either atom 3 or 4 should also be excited to satisfy the dimer constraint, which respectively yields $p_L \neq 0$ and $p_L = 0$. We can see that the number of dimer coverings with excitations on 1-3 and 1-4 is identical. Indeed, there exists a bijection $B$ between these dimer covering subsets [50]

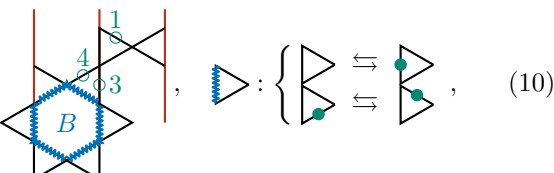

$$\text{(10)}$$

where we have defined $B$ by its action on each of the triangles. A similar analysis holds when atom 2 is in $|e\rangle$. As a result, there are as many dimer coverings leading to $p_L = 0$ and $p_L \neq 0$ once $p_R = 0$ has been measured. Since $|\Phi\rangle$ is an equal superposition of those, the probability of both outcomes is identical, equal to one half. If we write $K$ the probability to measure $p_R = 0$ during the first trial, the probability of failing to apply $G_\varphi$ on $n$ consecutive PEPS tensors is thus $K/2^{n-1}$, as claimed above.

Finally, notice that $\varphi$ can be tuned to realize, among others, the three gates $G_0 = Z$, $G_{\pi/2} = Z\sqrt{X}$ and $G_\pi = ZX$, which form a universal gate set for single qubit operations since $\sqrt{X}$ is a non-Clifford gate [51]. We have therefore demonstrated that it is possible to implement arbitrary single qubit operations along the computational wires of our system with a polynomial circuit depth.

### D. Universal quantum computation

To reach universality, the current gate set should be expanded with a two-qubit gate that is not unitarily equiv-

alent to a SWAP. We propose the following sequence, named $Q$: Three of bowtie's atoms are dephased by $\pi$ (III), for instance, the rightmost atoms and those next to the leftmost one; then pulse (II) is applied to the left and right triangles. This projects the bowtie's physical space onto

$$(\triangleright - p_\uparrow^L p_\downarrow^L \blacktriangleright + x^L p_\uparrow^L \blacktriangleright + x^L p_\downarrow^L \blacktriangleright)$$
$$\times (\triangleleft + p_\uparrow^R p_\downarrow^R \blacktriangleleft - x^R p_\uparrow^R \blacktriangleleft - x^R p_\downarrow^R \blacktriangleleft), \tag{11}$$

where the $x$, $p_\uparrow$ and $p_\downarrow$'s have been defined above. Using Eq. 5, one can check that the PEPS tensor obtained after contraction of the physical index acts on the two incoming logical qubits as

$$\text{(12)}$$

up to irrelevant Pauli errors [52]. Apart from the initial Hadamard gates common to all of our gate set, $Q$ can be understood as an entangling Bell gate that sends, for instance, $|00\rangle \rightarrow |01\rangle + |10\rangle$. Such a gate creates entanglement and therefore provides universality when appended to our previous single qubit operations.

## IV. CONCLUSION

Measurement-based quantum computation feeds from entangled many-body states, and can be realized if sufficient experimental control over each individual constituent of the may-body state exists. It is therefore particularly suited to arrays of Rydberg atoms, which have repeatedly demonstrated their strength as quantum simulators, and benefit from tried-and-tested laser manipulation techniques to measure all atoms individually.

In this article, we have shown that the entangled many-body state realized in Ref. [1] together with the proposed measurement pulses enable universal measurement-based quantum computation. This establishes Rydberg arrays as a new platform for universal quantum computing, which could potentially compete with superconducting quantum circuits due to the large number of available atoms (about 220 in Ref. [1]).

Our work calls for a search of robust and resilient Rydberg phases with identical computational power, over which single qubit operations could be performed with circuits of smaller depth. It is also important to explore how to identify, quantify and correct the intrinsic errors arising from the non-adiabatic state preparation in Rydberg arrays, and how to mitigate the effects of the long-range part of the van der Waals interaction through pulse engineering.

## V. ACKNOWLEDGMENTS

I gratefully acknowledges support from the MathWorks fellowship, and thank A. Wei and I. L. Chuang for valuable insights and comments.

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
