# Peer review of "Rydberg arrays offer a universal resource for measurement-based quantum computations"

_SciPost Physics_

## Round 1 · Referee Report · Anonymous (Referee 1) · 2022-6-24

Report

This paper proposed a measurement scheme for the Kagome lattice dimer state so that it can be used as a universal resource state for measurement-based quantum computation. It has recently been shown that the Kagome lattice dimer state can be approximately realized through a dynamical process in a Rydberg array. This work is hence timely and can be of interest to the community. On the other hand, the analysis presented in this paper is standard and does not involve much new idea / technique. The process of looking for a universal measurement scheme through the tensor network representation of a state is well known and has been carried out for many states. Moreover, the paper does not address the more serious issue of error / deviation in the resource state on the measurement scheme. Such analysis has been carried out for some resource state and the idea of a universal resource phase has been proposed. Given that, the result in this paper, from the theory perspective, is on the simple side. It in turn limits the impact this paper can have on real Rydberg experiments. I feel that the current version of the paper falls short of the originality criteria for publication in SciPost. To be considered for publication, the author needs to take the discussion to the next level.

---

## Editorial Decision

awaiting_resubmission